# PHF^3^ Technique: A Pyramid Hybrid Feature Fusion Framework for Severity Classification of Ulcerative Colitis Using Endoscopic Images

**DOI:** 10.3390/bioengineering9110632

**Published:** 2022-11-01

**Authors:** Jing Qi, Guangcong Ruan, Jia Liu, Yi Yang, Qian Cao, Yanling Wei, Yongjian Nian

**Affiliations:** 1Department of Digital Medicine, School of Biomedical Engineering and Imaging Medicine, Army Medical University (Third Military Medical University), Chongqing 400038, China; 2Department of Gastroenterology, Daping Hospital, Army Medical University (Third Military Medical University), Chongqing 400042, China; 3Department of Gastroenterology, Sir Run Run Shaw Hospital, Zhejiang University School of Medicine, Hangzhou 310016, China

**Keywords:** ulcerative colitis, Mayo endoscopic subscore, deep learning, hybrid architecture, feature fusion

## Abstract

Evaluating the severity of ulcerative colitis (UC) through the Mayo endoscopic subscore (MES) is crucial for understanding patient conditions and providing effective treatment. However, UC lesions present different characteristics in endoscopic images, exacerbating interclass similarities and intraclass differences in MES classification. In addition, inexperience and review fatigue in endoscopists introduces nontrivial challenges to the reliability and repeatability of MES evaluations. In this paper, we propose a pyramid hybrid feature fusion framework (${\mathrm{PHF}}^{3}$) as an auxiliary diagnostic tool for clinical UC severity classification. Specifically, the ${\mathrm{PHF}}^{3}$ model has a dual-branch hybrid architecture with ResNet50 and a pyramid vision Transformer (PvT), where the local features extracted by ResNet50 represent the relationship between the intestinal wall at the near-shot point and its depth, and the global representations modeled by the PvT capture similar information in the cross-section of the intestinal cavity. Furthermore, a feature fusion module (FFM) is designed to combine local features with global representations, while second-order pooling (SOP) is applied to enhance discriminative information in the classification process. The experimental results show that, compared with existing methods, the proposed ${\mathrm{PHF}}^{3}$ model has competitive performance. The area under the receiver operating characteristic curve (AUC) of MES 0, MES 1, MES 2, and MES 3 reached 0.996, 0.972, 0.967, and 0.990, respectively, and the overall accuracy reached 88.91%. Thus, our proposed method is valuable for developing an auxiliary assessment system for UC severity.

## 1. Introduction

Ulcerative colitis (UC) is a chronic inflammatory bowel disease characterized by mucosal inflammation, which begins in the rectum and extends proximally into the colon in a continuous manner. Bloody diarrhea is the most common early symptom of UC, and other clinical symptoms include abdominal pain, faecal urgency, tenesmus, and vomiting [1,2]. In recent years, although the incidence of UC has stabilized in developed regions, the disease burden remains high [3]. Moreover, in developing regions, with the acceleration of urbanization, the incidence of UC continues to increase, reaching 5.41 cases per 100,000 persons in India [4]. Endoscopy plays a fundamental role in the diagnosis, treatment, and management of UC, especially in monitoring disease activity and responses to treatment [5]. Endoscopic mucosal remission is an important therapeutic goal for UC, as well as the basis for evaluating future colorectal cancer risk and improving the prognostic quality of life [6]. Therefore, accurately assessing UC activity and the overall severity of the disease is critical for selecting the best management strategy for patients [7].

At present, the most commonly used evaluation index for assessing the severity of UC in clinical practice is the Mayo score [8], and the Mayo endoscopic subscore (MES) is the most important component of the overall Mayo score [6,9]. The MES evaluates the degree of damage to the intestinal mucosa. As shown in Figure 1, the MES classifies mucosal injury into four levels: normal or inactive, mild disease, moderate disease, and severe disease. However, the use of the MES for endoscopic evaluation is difficult and requires that endoscopists be trained. The reliance on subjective interpretations by endoscopists also hinders the reliability and repeatability of MES classification [10,11]. In addition, inexperience and review fatigue may lead endoscopists to misjudge the severity of UC, which may result in delayed treatment and missing the best time to change treatment decisions. However, artificial intelligence technology has been used to assist endoscopists in the rapid and accurate determination of UC severity classification.

In recent years, convolutional neural networks (CNNs) have made substantial progress in the field of computer vision and are widely used in medical image classification [12,13], segmentation, registration, reconstruction, and object detection [14,15,16]. Due to the powerful feature extraction ability of CNNs, CNN-based deep learning models have been applied to colonoscopies to identify a variety of diseases in the small intestine [17] and detect polyps [18], significantly reducing the workload of endoscopists. The excellent remote dependency modeling capabilities of the Vision Transformer (ViT) have popularized this approach in the field of computer vision [19]. Moreover, the number of technical reports on the ViT in medical image analysis has increased exponentially [20]. While the ViT can compensate for the CNN’s inability to capture global representations, the input mode of patch embedding ignores local details and lacks local inductive bias and an overall hierarchical structure. As a result, many excellent studies that combine the ViT and CNN to utilize their complementary advantages have emerged [21,22].

Compared with other medical imaging methods, colonoscopy images are closer to natural images and have three color channels. The progressive shooting characteristics of colonoscopy cause colonoscopy images to appear diversified, and many shooting points contain not only the features of the same intestinal lumen cross-section, but also the depth features of the intestinal cavity. In one colonoscopy image, although the upper left corner may be far from the lower right corner, the characteristics of these regions may be similar in the bowel lumen sectional space. Therefore, remote relationship modeling may be critical for extracting information from colonoscopy images. Inspired by the pyramid vision Transformer (PvT) proposed by Wang et al. [23,24] and rich CNN-Transformer feature aggregation networks [25], we propose a pyramid hybrid feature fusion framework (${\mathrm{PHF}}^{3}$) for UC severity classification. Compared with the ViT, the PvT can learn high-resolution representations while taking into account the problem of computational consumption and has a more nuanced local feature extraction process, but its relationship modeling is still global and lacks the inductive bias unique to convolution. The pyramidal hierarchical structure of the PvT creates a unique condition for its fusion with the CNN, which avoids the problem of feature dimension mismatch when the CNN is combined with Transformer. The design of the ${\mathrm{PHF}}^{3}$ dual-branch stream hybrid architecture ensures that the local features and the global representation are relatively independent while complementing each other, which provides a simple and effective way of feature fusion. The main contributions of this article can be summarized as follows:(1)The dual-branch pyramid hybrid architecture combines two feature extractors, namely a CNN and a PvT, to extract the deep features and cross-sectional spatial features of the intestinal cavity in colonoscopy images.(2)A feature fusion module was designed to integrate the local features extracted by the CNN and the global dependencies modeled by the PvT, thereby improving the classification accuracy by enhancing the feature representation ability.(3)At the output of the model, the second-order aggregation of the features was applied to enhance discriminative representations, which is effective for classifying the UC severity. In addition, an iterative method for covariance normalization was utilized to accelerate network training.

The remainder of this article is organized as follows. In Section 2, related work on UC severity classification and the hybrid architecture of the CNN and ViT is summarized. The proposed ${\mathrm{PHF}}^{3}$ model for UC severity classification is described in detail in Section 3. The performance of the ${\mathrm{PHF}}^{3}$ model is evaluated in Section 4 and compared with that of conventional deep models. A discussion and some conclusions are presented in Section 5 and Section 6, respectively.

## 2. Related Work

### 2.1. Deep Learning for UC Severity Classification

In recent years, deep-learning-based algorithms have replaced traditional machine learning methods due to superior recognition ability and end-to-end training strategies and have shown promise in gastroenteroscopy image diagnosis applications [26,27]. However, few studies have assessed the severity of UC. Ozawa et al. [28] constructed a computer-aided diagnosis system based on GoogleNet to identify MES 0 and MES 0-1, which was the first study that explored the performance of CNNs in evaluating different disease activity levels in UC. Subsequently, Stidham [29] showed that the deep learning model performed similarly to experienced human reviewers in grading the endoscopic severity of UC. In the recognition of MES 0-1 and MES 2-3, an Inception V3-based image classification architecture achieved an area under the receiver operation curve (AUC) of 0.970 (95% CI, 0.967–0.972). Bhambhvani et al. [6] developed a deep learning model based on ResNeXt101, with the aim of automatically classifying the MES of individuals with UC. However, this study included only three categories (MES 1-3), and the number of samples was small. Thus, recent deep-learning-based UC severity classification strategies are based on CNNs, and there have been few reports on four-level MES assessments. In a new study, Luo et al. [30] designed an efficient attention mechanism network (EAM-Net), and fed the features extracted from convolutional neural networks into EAM-Net and recurrent neural networks, respectively, achieving advanced results in the UC severity classification task, with an overall accuracy of 0.906 and 0.916 on two datasets, respectively. However, its DenseNet-based backbone may be difficult to complete the global modeling of the overall relationship [31].

### 2.2. Dual-Branch Stream Hybrid Architecture of CNN and ViT

Since the advantages and disadvantages of the CNN and ViT have been revealed, the combination of a CNN and ViT to develop a model with better performance has become a popular research topic. In general, these diverse works can be divided into three categories: conv-like Transformers, Transformer-like ConvNets, and conv-Transformer hybrids [32]. Among them, the conv-Transformer hybrid takes advantage of the CNN and ViT in a more direct and simpler way, and the dual-branch stream structure is one example of this type of model. In this kind of structure, an effective feature fusion module is critical. Peng et al. [31] proposed Conformer, which utilizes a convolution operation and a self-attention mechanism to enhance representation learning. The feature coupling unit of Conformer integrates local features and global representations at different resolutions in an interactive manner. Chen et al. [33] presented Mobile-Former, which adopts a parallel lightweight bidirectional bridge design between MobileNet and Transformer. Yoo et al. [25] designed a more concise feature aggregation method in which the flat features of Transformer linear embeddings are rearranged and concatenated and combined with CNN features. Due to the mismatch between the intermediate feature dimensions of the CNN and ViT, the design of the feature fusion module in these studies was relatively complex. Liu et al. [34] developed a hybrid architecture named CVM-Cervix, which does not include any interactions or fusion between the CNN and ViT branches, with a multilayer perceptron applied only at the output to combine the features of the two branches. The excellent performance of CVM-Cervix in cervical cancer classification tasks suggests that effective fusion at the output may be indispensable.

### 2.3. Higher-Order Statistics in Deep Learning

Since Lin et al. [35] proposed bilinear CNNs, many studies [36,37] have found that high-order pooling representations and deep CNN integration introduce promising improvements in challenging fine-grained visual classification tasks. Li et al. [37,38] conducted global covariance pooling for convolution features, achieving better improvements than those achieved by first-order pooling, and proposed a covariance iterative normalization method. Dai et al. [39], inspired by the work of Li et al., considered learning the feature interdependencies through the second-order statistics of the features and designed a second-order channel attention module for single-image super-resolution. Fang et al. [40] introduced a novel bilinear attention block for person retrieval, adopting the bilinear pooling method to model local feature interactions in each channel while preserving spatial structure information. Chen et al. [41] developed a new approach, fitting higher-order statistics with linear polynomials, and constructed a higher-order attention module for person re-identification, which can be simply realized by 1 × 1 convolution and an element-level addition/product. These encouraging studies demonstrate that higher-order statistics play a significant role in deep learning in enhancing the representations of discriminative features.

## 3. Materials and Methods

### 3.1. Dataset Details

This study was approved by the Ethics Committee of Daping Hospital affiliated with Army Medical University and was performed according to the Declaration of Helsinki. A total of 15,120 colonoscopy images with high quality of 768 cases were collected from the Daping Hospital affiliated with Army Medical University and Sir Run Run Shaw Hospital of Zhejiang University from January 2018 to December 2021. Each colonoscopy image was independently annotated by two endoscopic experts; when their labels were inconsistent, a third expert assisted in the discussion and they made the final decision together. Finally, the whole dataset included 4124 MES 0 images, 6669 MES 1 images, 1773 MES 2 images, and 2554 MES 3 images. Table 1 illustrates the specific data distribution. The whole dataset was randomly divided into training and test datasets at a ratio of 8:2, with the training dataset containing 12090 images and the test dataset containing 3030 images. Details on the datasets are presented in Table 2.

### 3.2. Overview of the Framework

The proposed ${\mathrm{PHF}}^{3}$ model is illustrated in Figure 2. The ${\mathrm{PHF}}^{3}$ model has a dual-branch structure, which consists of a PvT branch, a CNN branch, a feature fusion module (FFM), two dual-branch classifiers, and a fusion classifier based on second-order pooling (SOP). The structure of the PvT is described in detail in Section 3.2, while the CNN branch is based on the ResNet50 architecture. The PvT and ResNet50 model were both pretrained on the ImageNet dataset. In a colonoscopy image, the local features extracted by the CNN represent the relationship between the intestinal wall at the near-shot point and the coaxial extension line, while the global representations modeled by the PvT capture similar information in the cross-section space of the intestinal cavity. The FFM enhances the visual representation ability by combining local features with global representations in an interactive manner. The PvT and CNN branches both contain four stages, and the FFM performs feature fusion on the output from Stages 1 to 3. Then, the fused features are transmitted back to the two main branches. The ${\mathrm{PHF}}^{3}$ model has three outputs: the auxiliary outputs of the dual branches adopt average pooling, while the outputs of the fourth stage are concatenated and spliced at the channel level and serve as the main output of the model after SOP and the fully connected (FC) layers. Therefore, the total loss of the model includes the sum of three losses, $Loss_{all}$ = $\alpha Loss_{pvt}$ + $\beta Loss_{cnn}$ + $\gamma Loss_{combine}$, and the cross-entropy loss function with label smoothing is applied for all losses. α, β, and γ are the weight coefficients of the three losses, respectively, and their proportions are discussed in Section 4.1. Suppose that the true label corresponding to the *n*-th sample is $y_{n}\in \left\{1,2,\ldots ,K\right\}$, and $v=(v_{1},v_{2},\ldots v_{K})$ is the final output of the network, that is the prediction result of sample *n*. The calculation is expressed as follows:(1)$$\begin{array}{cc} & Loss_{\Omega }=\left(1-\varepsilon \right)\cdot \left[-\frac{1}{N}\sum_{n=1}^{N}\log(\frac{e^{v_{y_{n}}}}{\sum_{m=1}^{K}e^{v_{m}}})\right]+\varepsilon \cdot \left[-\frac{1}{NK}\sum_{n=1}^{N}\sum_{k=1}^{K}\log(\frac{e^{v_{k}}}{\sum_{m=1}^{K}e^{v_{m}}})\right] \end{array}$$
where *N* is the number of samples, *K* is the number of classification categories, and ε is the coefficient of label smoothing. Ω can stand for *pvt*, *cnn*, and *combine*.

### 3.3. Pyramid Vision Transformer

The PvT was originally proposed for dense prediction tasks, such as semantic segmentation and target detection, and is a pure Transformer backbone [23]. The progressive shrinking pyramid and spatial-reduction attention layer in the PvT can learn high-resolution representations, reducing the computational costs. Although the architecture of the PvT is favorable for dense prediction tasks, it does not show a strong advantage for image classification tasks. The improved version of the PvT [24] includes overlapping patch embedding (OPE) and a convolutional feed-forward network, thereby reducing the computational costs and exhibiting excellent performance in classification tasks. Each stage of the PvT consists of one OPE, one block, and one normalization layer, with the block containing N basic component blocks.

Figure 3 shows the components of the *i*-th stage of the PvT in our study. Let $H_{i}$, $W_{i}$, and $d_{i}$ be the height, width, and embedding dimensions of the features in the *i*-th stage, respectively. The flattened token output in the (*i*-1)-th stage is reshaped, and then, OPE is carried out. In contrast to linear embeddings in the ViT, OPE is realized mainly by convolution operations with a kernel size larger than the stride size. When *i* = 1, the convolution kernel size in OPE is 7 and the stride is 3, while when *i* = 2,3,4, the convolution kernel size in OPE is 3 and the stride is 2. In the basic component block, spatial reduction (SR) is performed first; then, multihead attention (MHA) is implemented. The MHA mechanism receives a query *Q*, key *K*, and value *V* as the input, and the SR operation greatly reduces the scale of *K* and *V*, effectively reducing the computational overhead and strongly encouraging the model to learn higher-resolution representations. The MHA operation can be formulated as follows:(2)$$\mathrm{MHA}\left(Q,K,V\right)=\mathrm{Concat}\left(head_{1},\cdots ,head_{h_{i}}\right)W^{O}$$
(3)$${\mathrm{head}}_{j}=\mathrm{Attention}\left(QW_{j}^{Q},\mathrm{SR}\left(K\right)W_{j}^{K},\mathrm{SR}\left(V\right)W_{j}^{V}\right)$$
where $h_{i}$ is the number of heads in the attention layer at stage *i* and Concat(·) is the concatenation operation. $W^{O}\in R^{d_{i}\times d_{i}}$ and $W_{j}^{Q},W_{j}^{K},W_{j}^{V}\in R^{d_{i}\times \left(d_{i}/h_{i}\right)}$ are linear projection parameters. The SR(·) operation can be formulated as follows:(4)$$\begin{array}{cc} & \mathrm{SR}\left(x\right)=\mathrm{LN}({\mathrm{RP}}_{2}\left(\mathrm{Conv}\left({\mathrm{RP}}_{1}\left(x\right),R_{s}^{i}\right)\right)W^{S}) \end{array}$$
where $x\in R^{H_{i}W_{i}\times d_{i}}$ denotes the input sequence and ${\mathrm{RP}}_{1}$(·) and ${\mathrm{RP}}_{2}$(·) are reshape operations. $x\in R^{d_{i}\times H_{i}\times W_{i}}$ denotes the feature after ${\mathrm{RP}}_{1}$(·), and $R_{s}^{i}$ represents the spatial reduction ratio, which is also the size of the kernel and stride in the convolution operation Conv(·).

At the end of the spatial reduction operation, ${\mathrm{RP}}_{2}$(·) changes $x\in R^{d_{i}\times \left(H_{i}/R_{s}^{i}\right)\times \left(W_{i}/R_{s}^{i}\right)}$ to $x\in R^{\left(H_{i}W_{i}/{\left(R_{s}^{i}\right)}^{2}\right)\times d_{i}}$ and LN(·) refers to layer normalization, while $W^{S}\in R^{d_{i}\times d_{i}}$ is a linear projection. Note that the Attention(·) calculation is consistent with the original paper [42]:(5)$$\begin{array}{cc} & \mathrm{Attention}\left(q,k,v\right)=\mathrm{Softmax}(\frac{qk^{⊤}}{\sqrt{d_{i}/h_{i}}})v \end{array}$$

Depthwise convolution is introduced into the convolution feed-forward network to capture the local continuity of the input tensor. The dimensional decay factor between the two fully connected (FC) layers at the *i*-th stage is $R_{m}^{i}$. In our study, the settings of *d*, *h*, $R_{s}$, and $R_{m}$ in the four stages were [64, 128, 320, 512], [1, 2, 5, 8], [8, 4, 2, 1], and [8, 8, 4, 4], while the numbers of basic component blocks in the four stages were 3, 8, 27, and 3.

### 3.4. Feature Fusion Module

As shown in Figure 2, we developed a complementary design between the two branches, namely the feature fusion module (FFM). The structure of the FFM is illustrated in Figure 4. The FFM receives feature maps from the PvT and CNN branches, and after the fusion at the channel scale, the output is sent back to the two main branches to enhance the complementary representation. Concretely, $F_{pvt}^{i}\in R^{C_{pvt}^{i}\times H_{i}\times W_{i}}$ and $F_{cnn}^{i}\in R^{C_{cnn}^{i}\times H_{i}\times W_{i}}$ are intermediate feature mappings from the PvT and CNN in stage *i*, which are aggregated by $G_{fuse}$:(6)$$\begin{array}{cc} & M_{pvt}^{i},M_{cnn}^{i}=\mathrm{Split}\left(G_{fuse}\left(\mathrm{Concat}\left(F_{pvt}^{i},F_{cnn}^{i}\right)\right)\right) \end{array}$$
where Concat(·) is the concatenation operation and Split(·) is the tensor split operation. $G_{fuse}$ consists of 1 × 1 convolutions and ReLU activation functions and is designed for channel-level fusion. The $M_{pvt}^{i}\in R^{C_{pvt}^{i}\times H_{i}\times W_{i}}$ and $M_{cnn}^{i}\in R^{C_{cnn}^{i}\times H_{i}\times W_{i}}$ obtained by splitting along the channel dimension are followed by $G_{pvt}$ and $G_{cnn}$. Then, the fused features are transmitted back to each branch and added to the original input features $F_{pvt}^{i}$ and $F_{cnn}^{i}$. It is worth noting that the FFM aggregates local features and global representations only in Stages 1 to 3; in Stage 4, the outputs of the two branches are concatenated, followed by SOP and the final classification.

### 3.5. Second-Order Pooling

In typical CNN structures, global average pooling implements first-order data statistics on the extracted features to determine the final classification. However, compared with the complex learning process of the CNN, the first-order statistics are relatively crude. Inspired by [37,38], in our study, second-order pooling was applied for the final abstraction of the features obtained by each branch. Specifically, $F_{pvt}^{4}\in R^{C_{pvt}^{4}\times H_{4}\times W_{4}}$ and $F_{cnn}^{4}\in R^{C_{cnn}^{4}\times H_{4}\times W_{4}}$ are the output features of the PvT and CNN branches in Stage 4, and $F_{final}\in R^{\left(C_{pvt}^{4}+C_{cnn}^{4}\right)\times H_{4}\times W_{4}}$ is obtained after the concatenation operation along the channel dimension. We reshaped $F_{final}$ to a feature matrix $X\in R^{C\times S}$, where $C=C_{pvt}^{4}+C_{cnn}^{4}$ and $S=H_{4}W_{4}$. The covariance matrix is calculated as follows:(7)$$\Sigma =X\bar{I}X^{⊤}$$
(8)$$\bar{I}=\frac{1}{S}\left(I-\frac{1}{S}1\right)$$

Here, **I** and **1** are identity and all-ones matrices of size *S* × *S*, respectively. Covariance normalization is beneficial for discriminative representations [39], and this normalization often relies on eigenvalue decomposition (EIG) or singular-value decomposition of the matrices. However, since graphics processing units (GPUs) are not ideal for EIG implementations, Newton–Schulz iterations were adopted to accelerate the covariance normalization process. To ensure the convergence of the Newton–Schulz iteration, Σ is first normalized as follows:(9)$$\begin{array}{cc} & A=\frac{1}{\mathrm{tr}(\Sigma )}\Sigma \end{array}$$

Here, $\mathrm{tr}\left(\Sigma \right)=\sum_{i}^{C}\lambda_{i}$ denotes the trace of Σ. Given $Y_{0}$ = **A** and $Z_{0}$ = **I**, for *l* = 1,…, ***L***, the Newton–Schulz iteration is given as follows:(10)$$\begin{array}{c} \begin{array}{c} Y_{l}=\frac{1}{2}Y_{l-1}\left(3I-Z_{l-1}Y_{l-1}\right)\\ Z_{l}=\frac{1}{2}\left(3I-Z_{l-1}Y_{l-1}\right)Z_{l-1} \end{array} \end{array}$$

Note that the pre-normalization process has an adverse effect on the network since it nontrivially changes the magnitude of the data. After the Newton–Schulz iteration, post-compensation is applied to produce the final normalized covariance matrix:(11)$$\begin{array}{cc} & \hat{Y}=\sqrt{\mathrm{tr}(\Sigma )}Y_{L} \end{array}$$
where $\hat{Y}$ is a symmetric matrix, and we extracted the upper triangular elements of this matrix to use as the input to the final fully connected layer.

### 3.6. Implementation Details

During training, randomly clipped 224 × 224 areas were fed into the network. During testing, the colonoscopy images were resized to 256 × 256, and center clipped 224 × 224 areas were used for prediction (for the InceptionV4 model, the image size was resized to 320 × 320 and the input size was 300 × 300). For data enhancement, random horizontal and vertical flips were adopted with a probability of 0.3. In addition, contrast-limited adaptive histogram equalization (Clahe) was applied to enhance the representation of the color features. The work of Mokter et al. [43] suggests that the vascular pattern is important characteristic information. As shown in Figure 5, the Clahe algorithm improved the image contrast and could highlight the vascular texture or white ulcers, while random flipping was conducive to increasing image diversity. The ${\mathrm{PHF}}^{3}$ model was trained using the stochastic gradient descent (SGD) optimizer with a weight decay of $1\times 10^{-5}$ and a momentum of 0.9. The learning rate was initialized to 0.001 and decreased by 0.1 every 10 epochs in the 50 training epochs, with 128 images included in each small batch. The cross-entropy loss with label smoothing was adapted, and the coefficient ε was set to 0.1. Our work was implemented using PyTorch with two NVIDIA Quadro GV100 32G graphics cards.

For the classification task, we evaluated the performance of all methods according to the overall accuracy and the accuracy (ACC), sensitivity (SEN), specificity (SPE), positive predictive value (PPV), negative predictive value (NPV), and F1-score (F1) of each individual class. These metrics can be calculated as follows:(12)$$total\;accuracy=TN_{all}/N$$
(13)accuracy=(TP+TN)/N
(14)sensitivity=TP/(TP+FN)
(15)specificity=TN/(TN+FP)
(16)positivepredictivevalue=TP/(TP+FP)
(17)negativepredictivevalue=TN/(TN+FN)
(18)$$F1-score=2\times \frac{recall\times precision}{recall+precision}$$

Here, *N* is the number of samples, while $TN_{all}$ denotes the number of all samples with correct predictions. *TP*, *TN*, *FP* and *FN* represent the number of true positive, true negative, false positive, and false negative samples in each prediction category, respectively.

## 4. Experiment and Results

### 4.1. Preliminary Study

In the final feature fusion process, the Newton–Schulz iterative procedure was applied to achieve fast covariance normalization, where the number of iterations was a tunable hyperparameter. Therefore, we first explored the impact of the number of Newton–Schulz iterations *L* on model performance. The results are shown in Figure 6, where *L* = 0 indicates that SOP was not adopted. When *L* = 1–8, the accuracy fluctuated to some extent, but the overall trend was rising. After *L* = 8, the overall accuracy decreased, indicating that increasing the number of iterations is not conducive to improving accuracy, which is consistent with the discussion in [37]. Therefore, in our final model, *L* was set to 8.

In addition, we also explored the proportion of weight coefficients α, β, and γ for the three losses: $Loss_{pvt}$, $Loss_{cnn}$, and $Loss_{combine}$. Intuitively, we believed that $Loss_{combine}$ was more important, and the experimental results are shown in Table 3. To a certain extent, increasing the weight of $Loss_{combine}$ can effectively improve the performance of the model. Therefore, we applied α:β:γ = 1:1:2 in our final model. On the training set, 5-fold cross-validation was applied to select various parameters for model training, and the area under the receiver operating characteristic curve (AUC) values of each class in the cross-validation analysis were 0.986, 0.951, 0.957, and 0.981 (Figure 7), while the overall accuracy was 85.45%.

### 4.2. The Ablation Experiments and Comparison Experiments with Classical CNNs

The results of the ablation experiments are shown in Table 4. Compared with the branch structure alone, the ${\mathrm{PHF}}^{3}$ model was able to combine the local features extracted by ResNet50 with the global representations modeled by the PvT to achieve better classification performance. The overall accuracy of ResNet50, the PvT, and the ${\mathrm{PHF}}^{3}$ model were 86.01%, 87.29%, and 88.91%, respectively. A performance comparison of some representative CNN models is shown in Table 5. Compared with ResNet50, ResNet101 had no significant improvement in terms of the recognition performance, suggesting that increasing the network depth did not enhance the representations of discriminant features. Considering the balance between performance and efficiency, our proposed ${\mathrm{PHF}}^{3}$ model adopted ResNet50 and the PvT as the two main branches and outperformed both individual models. The ROC curves of the five models on the test set are shown in Figure 8, and the AUC values of the ${\mathrm{PHF}}^{3}$ model for MES 0, MES 1, MES 2, and MES 3 reached 0.996, 0.972, and 0.990, respectively.

### 4.3. Comparative Experiments with Advanced Models

Even the basic ViT (ViT-B) exhibited better prediction accuracy than most CNNs. Moreover, compared to the ViT-B, the the PvT, which learns higher-resolution representations, improved the overall accuracy by approximately 1.0 percentage points and exhibited performance comparable to that of the basic Swin Transformer (Swin-B) [44]. In order to highlight the advantages of the ${\mathrm{PHF}}^{3}$ model, we also compared it with advanced models such as the VAN [45] and MViT [46], and Conformer [31], as a representative of the two-branch CNN-Transformer structure, was also included in our comparison scope. The results are shown in Table 6, from which we can find that the VAN did not achieve exciting performance. In comparison, the Swin-B , MViT, and PvT performed better, which implies that the effectiveness of the pyramid structure and the overlap and transformation of patches may be beneficial for learning diffuse lesions. The confusion matrices of the six models are shown in Figure 9, visually illustrating the differences in their predictions. Compared with the Swin-B, the ${\mathrm{PHF}}^{3}$ model had slightly higher false negatives for MES 1 and false positives for MES 3, but the Swin-B was more likely to predict MES 0 as MES 1 and MES 3 as MES 2. As a trade-off between recall and precision, the F1-score considers the false positive and false negative rates, and the ${\mathrm{PHF}}^{3}$ model had better F1-scores than the comparison models in all categories.

### 4.4. Visualization of Feature Maps and Heat Maps

The feature maps in the model inference process helped in understanding the feature capture characteristics of the CNN and PvT branches and the effectiveness of the FFM. As shown in Figure 10, we plotted partial feature maps before and after fusion for each stage. Overall, the extracted features of both branches became increasingly abstract as the model deepened, with the CNN branch focusing on local features and highlighting local details, while the PvT branch focused on global representations and the overall performance was chaotic. The FFM clearly introduced global information into the CNN branch, while the local details introduced into the PvT branch were more difficult to notice due to the chaotic feature representations. Furthermore, as shown in Figure 11, to enhance the interpretability of the model, Grad-CAM [47] was used to draw the heat maps. Some local ulcers and bleeding were highlighted, which are important features when the model makes decisions.

## 5. Discussion

### 5.1. A Novel Deep Learning Framework for UC Severity Assessment

In clinical practice, evaluating the severity of UC through MES is of great significance for understanding patient conditions and providing effective treatment. Ensuring the reliability and reproducibility of UC severity classification remains a nontrivial challenge, and previous works have been limited to convolutional neural networks, thus ignoring the global dependencies of features in the intestinal lumen. Our work provides a novel solution to this challenge. In colonoscopy images, we considered not only the local relationship between the depth features of the intestinal cavity, but also the global dependencies of the features in the same intestinal cavity cross-section. The proposed hybrid architecture combines local features and global representations in a simple and effective manner, achieving better performance than the baseline models. In the ablation study, the overall accuracy of the ${\mathrm{PHF}}^{3}$ model was 2.90 and 1.62 percentage points higher than that of ResNet50 and PvT, respectively. The feature fusion process of the two-branch hybrid framework can be observed obviously by the visualization of the feature maps. Compared with the classical CNNs, the accuracy of the proposed method was improved by 2.38–4.26 percentage points, and the AUC of the proposed method was the highest in all categories. Even compared to advanced models based on the Transformer architecture and CNN-Transformer combined framework, the ${\mathrm{PHF}}^{3}$ model still had the upper hand. Our approach highlighted the importance of the fusion of local features and global representations in the feature capture of diffuse lesions and the effectiveness of second-order information in enhancing fused discriminative features. This exciting result prompted us to believe that our study will further advance the application of deep learning as an auxiliary diagnostic tool in intestinal digestive diseases.

### 5.2. Multi-Branch Hybrid Architectures May Be Irreplaceable

The ViT and CNN are two mainstream models of deep learning at present. However, they all have their own shortcomings, such as the difficulty of global relationship modeling for the CNN and the lack of local inductive bias for the ViT. Therefore, a natural idea is to combine them to complement each other’s advantages, such as introducing convolution operators for the ViT or adding global attention mechanism for the CNN. Actually, the convolution operation is introduced into the PvT structure, which is a conv-like Transformer. Our experiments suggest that, even if convolutions are introduced to capture local relations in Transformers, the effects may be limited, and there is still room for improvement. Similarly, it is worth exploring whether Transformer-like ConvNets can be enhanced further by introducing the ViT. In studies combining the CNN and ViT, a hybrid structure with multiple branches may be a fusion method that is difficult to replace. Another issue worth noting is how to weight losses incurred by multiple branches. Although we tested weighting with different proportions, this artificial setup may not be optimal. If possible, designing adaptive weighting methods may yield surprising results. In addition, due to the complex attention mechanism and model design, the training and reasoning of the ViT are not as fast as those of the CNN, which increases the computational costs of the CNN-ViT fusion structure. The SR strategy in the PvT effectively reduced the amount of computations. In a recent report, inspired by the SR in the PvT, Li et al. [48] proposed the Next-ViT, a new paradigm that fuses convolutional and Transformer modules during every stage, aiming to improve model efficiency and achieve industrial-scale deployment of the CNN-Transformer hybrid architecture. Therefore, how to achieve efficient computation in a multi-branch hybrid framework is also a problem that needs more research and experiments.

### 5.3. Higher-Order Statistics Require More Exploration

It is well known that first-order statistics may limit modeling capabilities; however, second-order statistics are difficult to apply in GPUs because they introduce additional computations. In our study, covariance normalization realized by Newton–Schultz iterations achieved end-to-end training with acceptable computational costs, providing a new approach for higher-order statistics applications. In a preliminary study, we explored the influence of the number of Newton–Schultz iterations. Although the more iterations, the better the fitting, the experimental results showed that the existence of a certain fitting bias is beneficial to the generalization performance of the model. Our research may show that second-order information has higher discriminative representation power for the fusion features of two branches. A very smooth thought is whether statistics larger than second-order can perform better or whether it is an effective combination of first-order and second-order information that needs more research to explore. In addition, when the computational costs are no longer a hindrance, attention mechanisms based on higher-order statistics are an exciting concept. At present, some studies have used second-order statistical information to construct attention modules [39,40]. Furthermore, the kinds of feature representations that can be enhanced by higher-order information should be investigated.

### 5.4. Limitations and Future Work

Despite the excellent and encouraging classification results, our research has several limitations. First, the number of samples in each category in our dataset was not balanced, which is a characteristic of most clinical diseases: there are always more remission samples than severe case samples. UC severity classification is a more refined identification, and simple upsampling or downsampling of the data cannot provide effective improvements. Although class weighting based on loss can improve the classification accuracy of small sample categories, this process inevitably reduces the overall accuracy. Second, due to differences between the equipment and endoscopist manipulations, it is challenging to construct a model with strong generalizability. In our study, the data were obtained from only two large centers, and multicenter verification was lacking. In the future, we will consider collecting multisource data and applying our model to colonoscopy video processing. Moreover, we will explore the non-substitutability of the dual-branch hybrid architecture and applications of higher-order attention mechanisms.

## 6. Conclusions

In this paper, we proposed ${\mathrm{PHF}}^{3}$, a novel dual-branch hybrid feature fusion framework with the PvT and ResNet50 for UC severity classification in endoscopic images. Compared with the ViT, the PvT can learn higher-resolution representations, which is beneficial for learning the diffuse lesion features of UC. Moreover, the PvT has high operation efficiency. The designed FFM structure solves the feature fusion problem of the CNN and PvT as the feature resolution pyramid changes, thereby effectively combining local features and global representations, while the SOP module enhances discriminant information by using second-order statistics. The comparative ablation studies were encouraging, showing that the ${\mathrm{PHF}}^{3}$ model exhibited better performance than the comparison methods and can, thus, be used as an auxiliary tool for UC severity assessment. We hope that our work provides new ideas for combining convolution and Transformer and its application in assisted recognition in colonoscopy images.

## Figures and Tables

**Figure 1 bioengineering-09-00632-f001:**
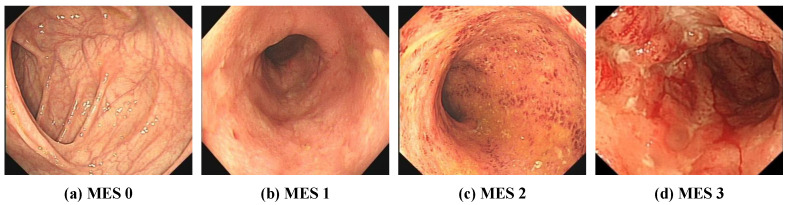
Representative images of Mayo endoscopy score. (**a**) MES 0: normal or inactive; (**b**) MES 1: mild, erythema, decreased vascular pattern, mild friability; (**c**) MES 2: moderate, marked erythema, absent vascular pattern, friability, erosions; (**d**) MES 3: severe, spontaneous bleeding, ulcerations.

**Figure 2 bioengineering-09-00632-f002:**
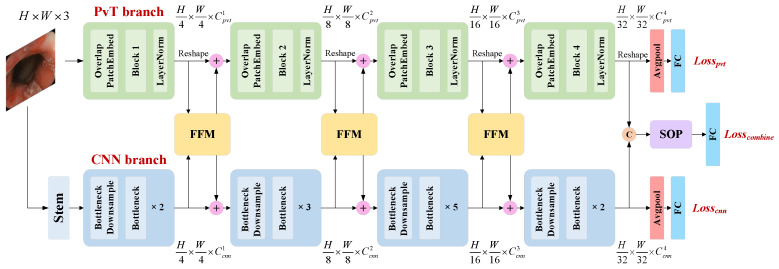
Illustration of the pyramid hybrid feature fusion framework (${\mathrm{PHF}}^{3}$). The stem consists of a convolution, a batch normalization, a ReLU activation function, and maximum pooling; FFM: feature fusion module, SOP: second-order pooling, LayerNorm: layer normalization, FC: fully connected layer.

**Figure 3 bioengineering-09-00632-f003:**
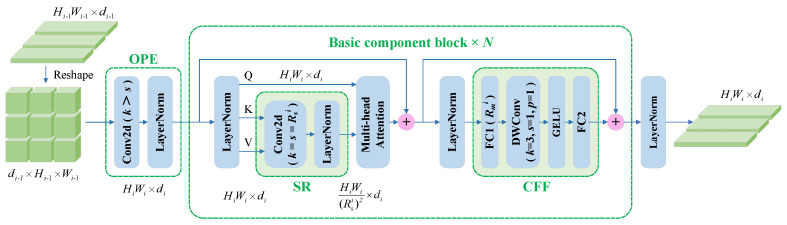
The basic component block of the pyramid vision Transformer (PvT). OPE: overlapping patch embedding, SR: spatial reduction, CFF: convolutional feed-forward.

**Figure 4 bioengineering-09-00632-f004:**
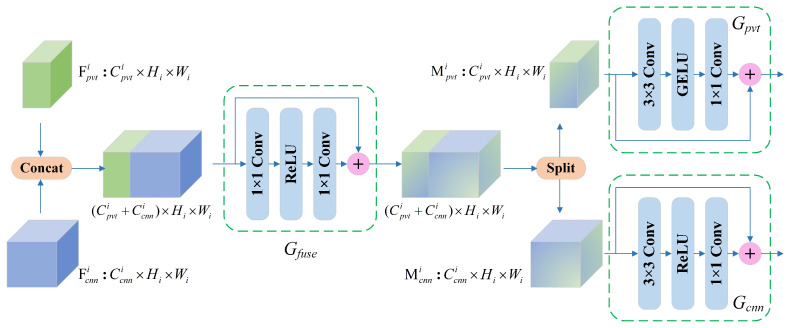
The structure of the feature fusion module (FFM).

**Figure 5 bioengineering-09-00632-f005:**
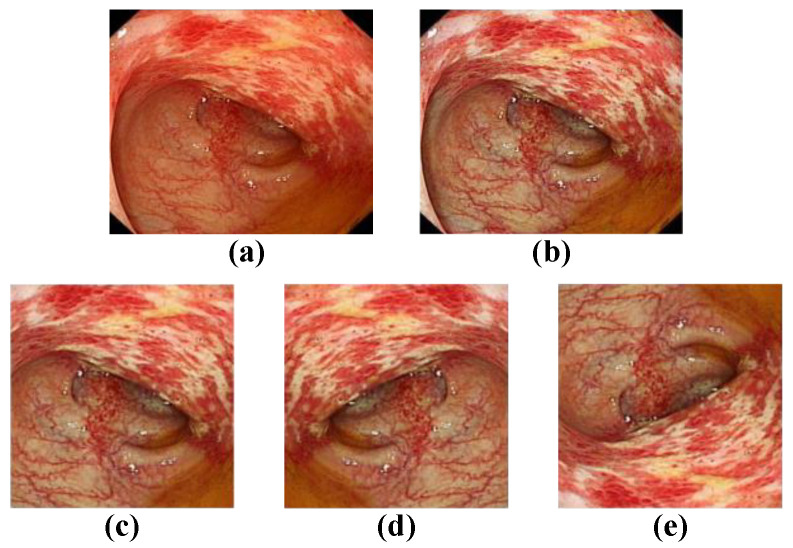
Data enhancement presentation. (**a**) The original image; (**b**) contrast-limited adaptive histogram equalization (Clahe); (**c**) image clipping; (**d**) horizontal flipping; (**e**) vertical flipping.

**Figure 6 bioengineering-09-00632-f006:**
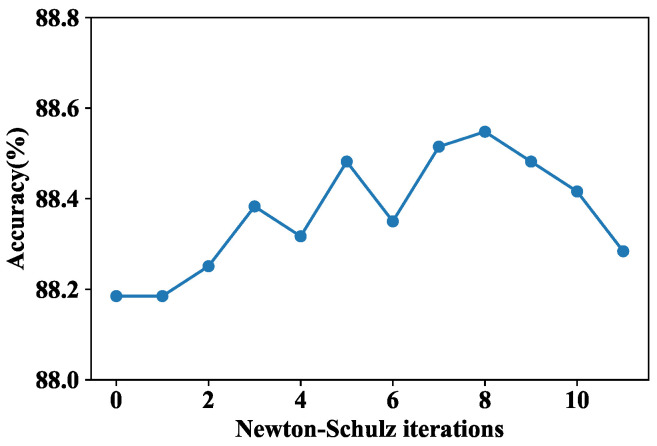
Performance comparison of ${\mathrm{PHF}}^{3}$ under different Newton–Schulz iterations *L*.

**Figure 7 bioengineering-09-00632-f007:**
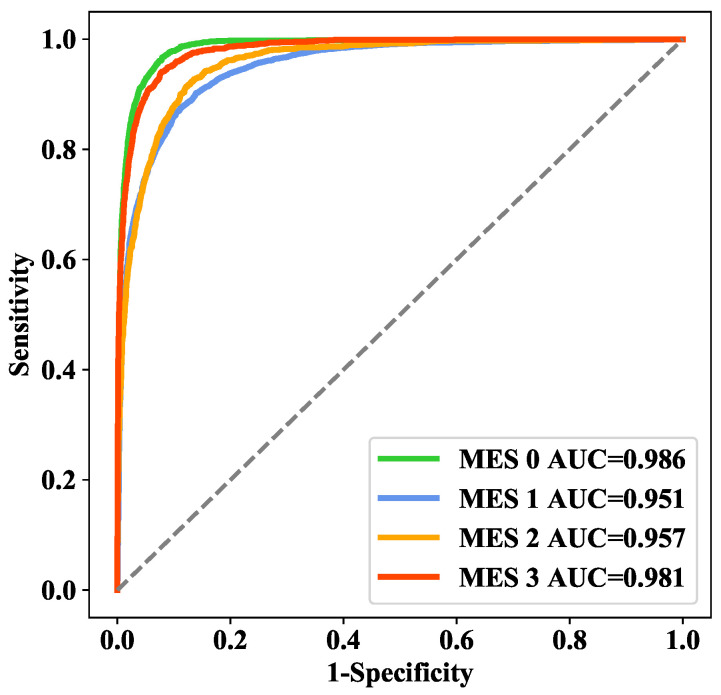
ROC curves for 5-fold cross-validation.

**Figure 8 bioengineering-09-00632-f008:**
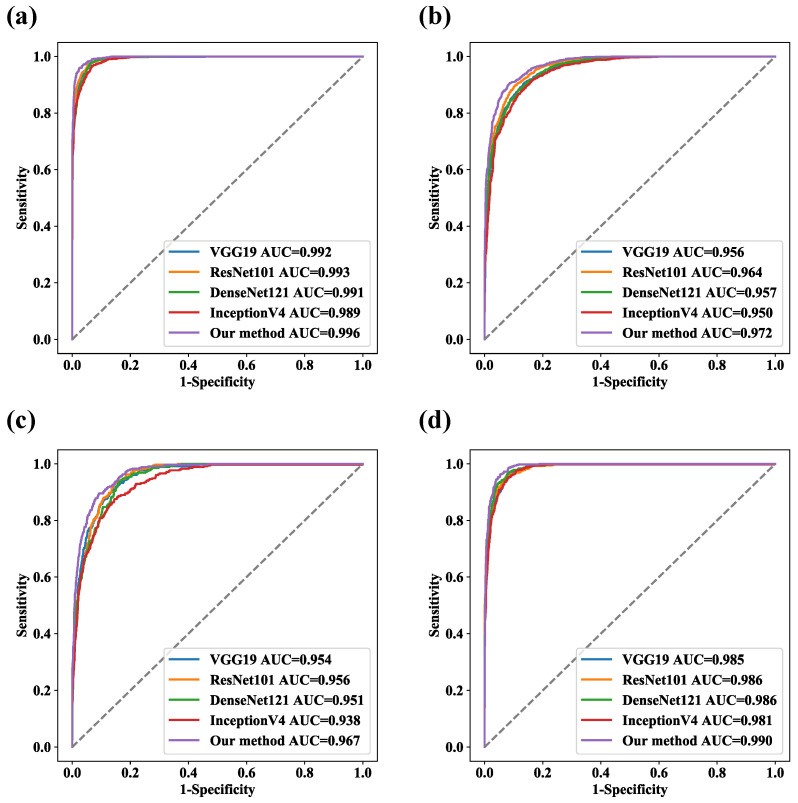
ROC curves for five models. (**a**) MES 0, (**b**) MES 1, (**c**) MES 2, and (**d**) MES 3.

**Figure 9 bioengineering-09-00632-f009:**
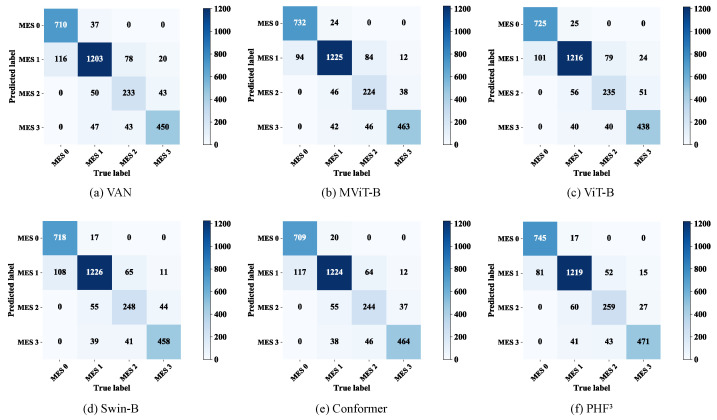
Confusion matrix for the six models.

**Figure 10 bioengineering-09-00632-f010:**
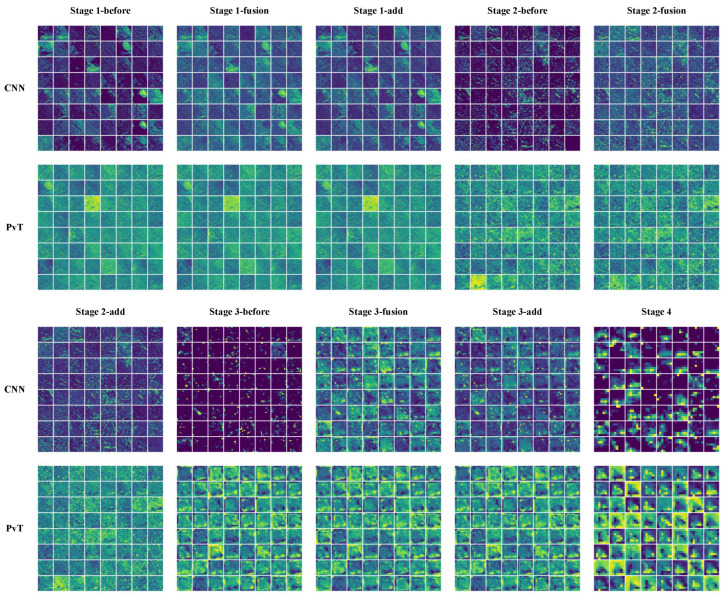
Partial feature maps before and after fusion at each stage. before: feature maps before fusion, fusion: feature maps after fusion, add: the blended features are superimposed on the original image and are also the input for the next stage.

**Figure 11 bioengineering-09-00632-f011:**
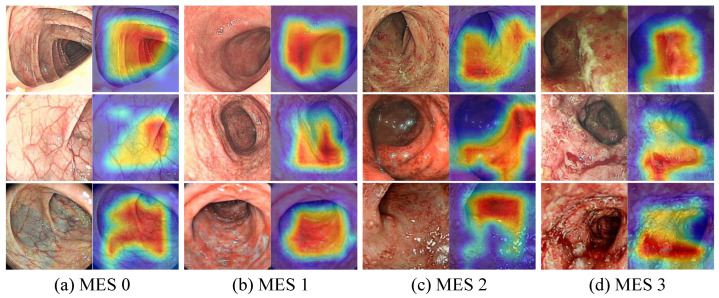
Visualization of the attention maps obtained by Grad-CAM.

**Table 1 bioengineering-09-00632-t001:** The specific data distribution.

	MES 0	MES 1	MES 2	MES 3	Total
Daping Hospital	3299	2479	950	1010	7738
Sir Run Run Shaw Hospital	825	4190	823	1544	7382
Total	4124	6669	1773	2554	15,120

**Table 2 bioengineering-09-00632-t002:** Distribution of experimental datasets.

	MES 0	MES 1	MES 2	MES 3	Total
Training dataset	3298	5332	1419	2041	12,090
Test dataset	826	1337	354	513	3030
Total	4124	6669	1773	2554	15,120

**Table 3 bioengineering-09-00632-t003:** Exploration of the ratio of α, β, and γ.

α:β:γ	1:1:1	1:1:2	1:1:3	1:1:4	1:1:5
Total ACC	88.55	88.91	88.78	88.45	88.05

**Table 4 bioengineering-09-00632-t004:** The ablation study.

Model	Metrics	MES 0	MES 1	MES 2	MES 3
ResNet50	ACC	95.48	89.04	93.00	94.49
	SEN	89.59	88.63	62.71	89.47
	SPE	97.69	89.37	**97.01**	95.51
	PPV	93.55	86.81	73.51	80.24
	NPV	96.16	90.87	95.16	97.80
	F1	91.53	87.71	67.68	84.61
PvT	ACC	95.81	90.17	93.37	95.25
	SEN	88.14	**91.32**	69.77	87.52
	SPE	98.68	89.25	96.49	**96.82**
	PPV	96.17	87.03	72.43	**84.88**
	NPV	95.69	92.87	96.02	97.44
	F1	91.98	89.12	71.08	86.18
${\mathrm{PHF}}^{3}$	ACC	**96.77**	**91.22**	**93.99**	**95.84**
	SEN	**90.19**	91.17	**73.16**	**91.81**
	SPE	**99.23**	**91.26**	96.75	96.66
	PPV	**97.77**	**89.17**	**74.86**	84.86
	NPV	**96.43**	**92.90**	**96.46**	**98.30**
	F1	**93.83**	**90.16**	**74.00**	**88.20**

**Table 5 bioengineering-09-00632-t005:** Comparison of the proposed ${\mathrm{PHF}}^{3}$ with some representative CNN models.

	VGG19	ResNet101	DenseNet121	InceptionV4	${\mathrm{PHF}}^{3}$
**Total ACC**	**85.81**	**86.53**	**85.45**	**84.65**	**88.91**
MES 0					
ACC	95.48	95.84	94.88	94.65	**96.77**
SEN	89.23	88.98	87.29	88.26	**90.19**
SPE	97.82	98.41	97.73	97.05	**99.23**
PPV	93.89	95.45	93.51	91.81	**97.77**
NPV	96.04	95.97	95.35	95.66	**96.43**
F1	91.50	92.11	90.29	90.00	**93.83**
MES 1					
ACC	88.75	89.70	88.25	87.62	**91.22**
SEN	86.91	89.90	90.28	88.11	**91.17**
SPE	90.19	89.55	86.65	87.24	**91.26**
PPV	87.50	87.16	84.23	84.51	**89.17**
NPV	89.72	91.82	91.86	90.28	**92.90**
F1	87.20	88.51	87.15	86.27	**90.16**
MES 2					
ACC	93.00	92.48	92.67	92.48	**93.99**
SEN	69.77	68.08	59.60	60.73	**73.16**
SPE	96.08	95.70	**97.05**	96.67	96.75
PPV	70.17	67.70	72.76	70.72	**74.86**
NPV	96.00	95.77	94.78	94.90	**96.46**
F1	69.97	67.89	65.53	65.35	**74.00**
MES 3					
ACC	94.39	95.05	95.08	94.55	**95.84**
SEN	88.50	86.55	87.72	86.35	**91.81**
SPE	95.59	**96.78**	96.58	96.23	96.66
PPV	80.35	84.57	83.96	82.34	**84.86**
NPV	97.61	97.25	97.47	97.19	**98.30**
F1	84.23	85.55	85.80	84.30	**88.20**

**Table 6 bioengineering-09-00632-t006:** Comparison of the proposed ${\mathrm{PHF}}^{3}$ with some advanced Transformer models.

	VAN [45]	MViT-B [46]	ViT-B [42]	Swin-B [44]	Conformer [31]	${\mathrm{PHF}}^{3}$
**Total ACC**	**85.68**	**87.26**	**86.27**	**87.46**	**87.16**	**88.91**
MES 0						
ACC	94.95	96.11	95.84	95.87	95.48	**96.77**
SEN	85.96	88.62	87.77	86.92	85.84	**90.19**
SPE	98.32	98.91	98.87	**99.23**	99.09	**99.23**
PPV	95.05	96.83	96.67	97.69	97.26	**97.77**
NPV	94.92	95.87	95.57	95.29	94.92	**96.43**
F1	90.27	92.54	92.01	91.99	91.19	**93.83**
MES 1						
ACC	88.51	90.03	89.27	90.26	89.90	**91.22**
SEN	89.98	91.62	90.95	**91.70**	91.55	91.17
SPE	87.36	88.78	87.95	89.13	88.60	**91.26**
PPV	84.90	86.57	85.63	86.95	86.38	**89.17**
NPV	91.69	93.07	92.48	**93.15**	92.99	92.90
F1	87.36	89.03	88.21	89.26	88.89	**90.16**
MES 2						
ACC	92.94	92.94	92.54	93.23	93.33	**93.99**
SEN	65.82	63.28	66.38	70.06	68.93	**73.16**
SPE	96.52	**96.86**	96.00	96.30	96.56	96.75
PPV	71.47	72.73	68.71	71.47	72.62	**74.86**
NPV	95.53	95.22	95.57	96.05	95.92	**96.46**
F1	68.53	67.67	67.53	70.76	70.72	**74.00**
MES 3						
ACC	94.95	95.45	94.88	95.54	95.61	**95.84**
SEN	87.72	90.25	85.38	89.28	90.45	**91.81**
SPE	96.42	96.50	**96.82**	**96.82**	96.66	96.66
PPV	83.33	84.03	84.56	**85.13**	84.67	84.86
NPV	97.47	97.98	97.01	97.79	98.03	**98.30**
F1	85.47	87.03	84.97	87.16	87.46	**88.20**

## Data Availability

The data cannot be made available due to privacy restrictions.

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
