# Peer review of "PHF3 Technique: A Pyramid Hybrid Feature Fusion Framework for Severity Classification of Ulcerative Colitis Using Endoscopic Images"

_bioengineering, 2022, doi:10.3390/bioengineering9110632_

Round 1

Reviewer 1 Report

1. General comments:

The paper presents a pyramid hybrid feature fusion framework as an auxiliary diagnostic tool for clinical UC severity classification. The proposed model has a dual-branch hybrid architecture with ResNet50 and a pyramid vision transformer. The local features extracted by ResNet50 express the relationship between the intestinal wall at the near shot point  and its depth, and the global representations modelled by the PvT capture similar information in the cross-section of the intestinal cavity. The authors also designed a feature fusion module to combine local features with global representations, while second-order pooling is applied to enhance discriminative information in the classification process.

2. Analysis:

Quality of the writing: structured coherently

Abstract: Easy to understand

Introduction: well-written, context is clear to some extent.

Problematic: clearly-identified

Method: well-described

Application field: well-identified

Results: well-illustrated.

Conclusion: concise

Related works: more recent references are required.

Originality: yes

Recommendations: Add more recent references (2020, 2021, 2022), including:

Classification of Ulcerative Colitis Severity

1.     M.F. Mokter et al. (2020), “Classification of Ulcerative Colitis Severity in Colonoscopy Videos Using Vascular Pattern Detection”, Machine Learning in Medical Imaging, 2020, https://link.springer.com/chapter/10.1007/978-3-030-59861-7_56

2.     C. Pagnini et al. (2021), “Mayo Endoscopic Score and Ulcerative Colitis Endoscopic Index Are Equally Effective for Endoscopic Activity Evaluation in Ulcerative Colitis Patients in a Real Life Setting”, Gastroenterology Insights, 2021, https://www.mdpi.com/2036-7422/12/2/19

Deep Learning methods applied to detection and classification of diseases.

3.     A.K. Dutta et al. (2022) “Barnacles Mating Optimizer with Deep Transfer Learning Enabled Biomedical Malaria Parasite Detection and Classification”, Computational Intelligence and Neuroscience (Q1), 2022, 1-12; https://doi.org/10.1155/2022/7776319

4.     A.Raza et al. (2022) “A Hybrid Deep Learning-Based Approach for Brain Tumor Classification”, Electronics, 2022; doi.org/10.3390/electronics11071126

Reviewer 2 Report

Manuscript Title “PHF3: A pyramid hybrid feature fusion framework for severity classification of ulcerative colitis using endoscopic images”

General comment:

the quality of this manuscript is pretty good and rigid actually, but the structure of the manuscript needs to be totally revised again to fulfill the format of journal article. The content of an article usually categorized into introduction, materials and methods, results, and discussion, however the specific one has a weird organization and makes it very difficult to understand.

Specific comment:

1.      Title: add technique after PHF3 as PHF3 technique…… might be better for understanding

2.      Abstract: ok, the experimental process and the solid results are all mentioned herein.

3.      Introduction: included the related work should be merged altogether to become one part of introduction with several sections. The length needs to be shortened too, since the rationale study is always not the topic in the article, just long enough to lead the reader into the correlated topic and propose the theoretical basis of the method in enough

4.      Materials and methods: figure 2 is redundant, suggested to be replaced by a table for saving space, plus the test data set is too few to verify the training process, usually it is 4:1 for training to test group.

5.      Experiment and results: the definition of the chapter is unsuitable, suggest to add some sub-section heads for easy recognizing the topics of statement.

6.      4.1 should be the preliminary study, 4.2 test result or author has any solid suggestion?

7.      Discussion: the content in discussion is too few to illustrate the technique. Usually, it occupied 1/5-1/3 total length of the manuscript, plus, no sub-section head to define the topic of statement.

Reviewer 3 Report

In this study, the authors introduced a new method that employed a UC severity classification. A pyramid hybrid feature fusion framework is proposed.

My major concerns related to this work are as detailed below:

1.  Use in the figures the same abbreviations as in the main text (ex. OPE)

2.     Please provide the size of the kernel used in OPE

3.     Figs. 3, 4 and 5 please clarify the size of the feature space. There are some inconsistencies in your notations

4.     Eq. (4): please provide the meaning of the used quantities and their equivalence with your work

5.     The details of loss function are missing.

Reviewer 4 Report

This study developed a new algorithm attempting to leverage the local and global features by fusing CNN and transformer. The new model was evaluated in differentiating four stages of ulcerative colitis, with performance comparison to various CNN and transformer algorithms. This is a welcomed exploration aiming to make the most out of the latest algorithms and apply their new model to a classification problem of clinical interest. This study is overall well planned and well conducted. The presentation is also well written and easy to follow. The results are presented in a logical way, with a clear description of their observations and explanations of underlying reasons. I read through the manuscript and haven’t found typos or grammar mistakes. Congratulations! A well-done job.    

Reviewer 5 Report

The authors conducted a retrospective study testing their hybrid feature-fusion framework in classifying the severity of patients with ulcerative colitis. There are some major questions that need to be answered before this manuscript is considered publishable. 

It is unclear why the pyramid vision transformer (PvT) is selected for this hybrid feature-fusion framework task. Since the authors described many downsides of applying PvT for the image classification task, it is unclear why we need all the tasks to adapt this transformer for severity classification in this manuscript. In addition, this complicated framework only performed marginally better than other more straightforward frameworks (Table 3) 

The authors have an average loss of three, representing the loss from PvT, CNN, and the SOP branches. It does not seem correct to have the same weight for all three branches since the combined outcome should be more important than the other two.

Minor: 

Line 114 However, its DenseNet-based backbone may

be difficult to complete the global modeling of the overall relationship.

Citation and explanation needed. 

Line 104: "recognition of MES 0-1 and MES 2-3, an Inception V3 based image classification architecture achieved an area under the receiver operation curve (AUC) of 0.966."

Clearly cite the literature. Give the confidence interval of the AUC to show the precision of this small study. 

Line 113: results in the UC severity classification task

Results Should be given. 

The ratio of testing vs. training in categories MES 0-3 are: 1: 0.14, 0.11, 0.14, and 0.13, which is quite unusual given the authors' description in their methods. This might also influence the different performances of the models. 

The authors should describe the pre-processing procedure, including image clipping, flipping, and contrast equalization in the methods. 

The ablation experiments were not clearly defined. If these are the results from the testing dataset, it is not clear why the model performed better in the testing dataset (Table 2 and Figure 8)

Round 2

Reviewer 2 Report

the revised version is largely elaborated the figure captions and also the newly added sub-section heads in either reesult or discussion. the quality right now can fulfill the academic criteria of the specific journal. therefore, I suggest to accept the manuscript in this format!

Reviewer 3 Report

Authors have incorporated all the suggestions made. The paper is in much better shape now and may be accepted for publication.

Reviewer 5 Report

All comments have been addressed satisfactorily. The manuscript was significantly improved and could be accepted in its present form.